# Integration of Spatial PD-L1 Expression with the Tumor Immune Microenvironment Outperforms Standard PD-L1 Scoring in Outcome Prediction of Urothelial Cancer Patients

**DOI:** 10.3390/cancers13102327

**Published:** 2021-05-12

**Authors:** Veronika Weyerer, Pamela L. Strissel, Reiner Strick, Danijel Sikic, Carol I. Geppert, Simone Bertz, Fabienne Lange, Helge Taubert, Sven Wach, Johannes Breyer, Christian Bolenz, Philipp Erben, Bernd J. Schmitz-Draeger, Bernd Wullich, Arndt Hartmann, Markus Eckstein

**Affiliations:** 1Institute of Pathology, University Hospital Erlangen, Friedrich-Alexander-Universität Erlangen-Nürnberg, Krankenhausstr. 8-10, 91054 Erlangen, Germany; veronika.weyerer@uk-erlangen.de (V.W.); strisspa@outlook.com (P.L.S.); carol.geppert@uk-erlangen.de (C.I.G.); simone.bertz@uk-erlangen.de (S.B.); fabienne.lange@uk-erlangen.de (F.L.); arndt.hartmann@uk-erlangen.de (A.H.); 2Comprehensive Cancer Center Erlangen-EMN (CCC ER-EMN), 91054 Erlangen, Germany; reiner.strick@uk-erlangen.de (R.S.); danijel.sikic@uk-erlangen.de (D.S.); helge.taubert@uk-erlangen.de (H.T.); sven.wach@uk-erlangen.de (S.W.); Bernd.Wullich@uk-erlangen.de (B.W.); 3BRIDGE-Consortium Germany e.V., 68177 Mannheim, Germany; johannes.breyer@ukr.de (J.B.); Christian.Bolenz@uniklinik-ulm.de (C.B.); philipp.erben@medma.uni-heidelberg.de (P.E.); bernd_sd@yahoo.de (B.J.S.-D.); 4Department of Gynecology and Obstetrics, University Hospital Erlangen, Friedrich-Alexander-Universität Erlangen-Nürnberg, 91054 Erlangen, Germany; 5Adjunct Affiliation With Department of Radiation Oncology, University of Maryland School of Medicine, Baltimore, MD 21201, USA; 6Department of Urology and Pediatric Urology, University Hospital Erlangen, Friedrich-Alexander-Universität Erlangen-Nürnberg, 91054 Erlangen, Germany; 7Department of Urology, University Hospital Regensburg, University of Regensburg, 93053 Regensburg, Germany; 8Department of Urology and Pediatric Urology, University of Ulm, 89081 Ulm, Germany; 9Department of Urology, University Hospital Mannheim, Rupprecht-Karls-Universität Heidelberg, 68177 Mannheim, Germany; 10Urologie 24, 90431 Nuremberg, Germany

**Keywords:** bladder cancer, urothelial cancer, immune phenotypes, PD-L1, PD-1, TILs

## Abstract

**Simple Summary:**

Diagnostic PD-L1 assessment of urothelial cancer to predict a patient’s immune therapy response remains a matter of controversy. Several contributing factors have been discussed; however, systematic studies are lacking. The present study demonstrates that clinically applied PD-L1 scoring algorithms are influenced by inter-algorithm variability and result in the selection of different “PD-L1” positive populations within the tumor immune microenvironment (TIME). The results further demonstrate that specific immune phenotypes of muscle-invasive urothelial cancer are associated with very different clinical outcomes, which cannot be resolved by PD-L1 testing. Thus, PD-L1 alone not only fails to reflect the TIME, but also has implications for patients. We conclude that a comprehensive integration of PD-L1 expression and immune phenotypes is superior to PD-L1 testing. This might be a novel strategy to predict a patient’s response to immune therapy.

**Abstract:**

Background: Immune therapy has gained significant importance in managing urothelial cancer. The value of PD-L1 remains a matter of controversy, thus requiring an in-depth analysis of its biological and clinical relevance. Methods: A total of 193 tumors of muscle-invasive bladder cancer patients (MIBC) were assessed with four PD-L1 assays. PD-L1 scoring results were correlated with data from a comprehensive digital-spatial immune-profiling panel using descriptive statistics, hierarchical clustering and uni-/multivariable survival analyses. Results: PD-L1 scoring algorithms are heterogeneous (agreements from 63.1% to 87.7%), and stems from different constellations of immune and tumor cells (IC/TC). While Ventana IC5% algorithm identifies tumors with high inflammation and favorable baseline prognosis, CPS10 and the TCarea25%/ICarea25% algorithm identify tumors with TC and IC expression. Spatially organized immune phenotypes, which correlate either with high PD-L1 IC expression and favorable prognosis or constitutive PD-L1 TC expression and poor baseline prognosis, cannot be resolved properly by PD-L1 algorithms. PD-L1 negative tumors with relevant immune infiltration can be detected by sTILs scoring on HE slides and digital CD8^+^ scoring. Conclusions: Contemporary PD-L1 scoring algorithms are not sufficient to resolve spatially distributed MIBC immune phenotypes and their clinical implications. A more comprehensive view of immune phenotypes along with the integration of spatial PD-L1 expression on IC and TC is necessary in order to stratify patients for ICI.

## 1. Introduction

Muscle-invasive bladder cancer (MIBC) shows a life-threatening clinical behavior with frequent metastatic spread and poor 5-year survival rates [1,2]. Standard treatment includes radical cystectomy and perioperative platinum-based chemotherapy, while systemic palliative treatment for patients with locally non-resectable and/or metastasized urothelial bladder cancer (mUC) consists of different chemotherapy regimens [3]. The approval of immune checkpoint inhibitors (ICI) targeting the protein cell death protein 1 (PD-1) and its ligand protein cell death ligand 1 (PD-L1) fundamentally changed the first- and second-line treatment options for mUC patients with the potential to induce long-term responses [4,5,6,7,8,9,10]. Ongoing trials (e.g., NCT03924856, NCT03732677, NCT03661320, NCT03294304) will hopefully confirm the first encouraging results showing high efficacy of ICI in neoadjuvant/perioperative settings.

PD-1 and PD-L1 play a critical role in the natural regulatory interplay of immune cells (IC), but are deregulated in tumor cells (TC), leading to tumor-mediated immune suppression [11]. PD-L1 is strongly linked with the inflammatory response and represents one of many interferon stimulated genes (ISGs) [12,13]. PD-L1 assessment on IC and TC emerged as a useful tool to predict ICI response in non-small cell lung cancer and head and neck squamous cell carcinoma [14,15]. However, in mUC, predictive and prognostic value of PD-L1 is inconsistent between therapy lines (front-line vs. second-line) and different drugs (e.g., atezolizumab vs. pembrolizumab) [4,5,6,7,8,9,10,16,17]. Based on survival data of two phase III trials in patients with mUC receiving first-line atezolizumab/pembrolizumab monotherapy, ICI was restricted to patients with PD-L1 positive tumors in 2018 (unpublished data for pembrolizumab) [16,17]. Approved PD-L1 companion diagnostic assays (CDA) show broad analytical concordance for IC and TC detection, with the exception of a decreased sensitivity using the Ventana SP142 assay for TC detection [18,19,20,21,22]. The main variance of PD-L1 scoring stems from different drug-specific scoring algorithms, which are based on completely different assessment methodologies for IC, TC, or both, and different cut-offs for PD-L1 positivity (e.g., pembrolizumab: combined positive score, CPS, cut-off = 10%; atezolizumab: Ventana IC score, cut-off = 5%; durvalumab: TC_area_/IC_area_ algorithm, cut-off = 25%) [18,23]. Differences in scoring algorithms and cut-off systems have been considered to substantially contribute to the inconsistent predictive and prognostic value of PD-L1 across different trials and ICI drugs [5,7,24]. It is important to note that a systematic analysis of inter-algorithm comparability along with the immunological context of the MIBC tumor immune microenvironment (TIME) is lacking. Although PD-L1 is currently used as a biomarker in daily practice to predict responses to immunotherapy, little is known about the intrinsic meaning of differential PD-L1 expression on TC and IC for immune therapy-naïve patients and if these patients would respond differently following immunotherapy or not. The investigation of differential PD-L1 expression in the MIBC TIME is of enormous importance, especially based on our initial findings, which indicate that different spatially organized immune phenotypes exist, where strong aberrant PD-L1 expression on TC is associated with a poor baseline prognosis, while patients with predominant PD-L1 expression on IC have an excellent prognosis [25]. These findings support differential PD-L1 regulation in the MIBC TIME. However, the extent to which current clinically applied PD-L1 scoring algorithms can correctly predict these immune inflammatory phenotypes with opposite prognostic and predictive outcomes has not yet been investigated.

We present comprehensive data deciphering immunological correlates of clinically applied PD-L1 scoring algorithms in 193 immunotherapy-naïve, consecutively recruited MIBC patients. We show that PD-L1 scoring algorithms correlate with different inflammation levels and immune cell phenotypes where PD-L1 expression distributes differentially among IC and TC, resulting in diverse patient outcomes. Furthermore, we demonstrate that the scoring of stromal tumor-infiltrating lymphocytes (sTILs) and CD8^+^ cytotoxic T-lymphocytes is an important tool to identify inflamed tumors classified as PD-L1 negative.

## 2. Materials and Methods

### 2.1. Study Cohort and Ethical Aspects

We studied 193 MIBC cases of the Comprehensive Cancer Center Erlangen Metropol Region Nuremberg (CCC-EMN) cohort treated by radical cystectomy, bilateral lymph node dissection, and adjuvant platinum-containing chemotherapy between 2004 and 2016 [18,25,26]. All cases were selected in accordance with current REMARK guidelines. Cases were systematically re-reviewed by two uropathologists (A.H., M.E.) according to the UICC TNM staging manual (8th edition) and the WHO 2016 classification of genitourinary tumors [27]. Clinico-pathologic data are shown in Appendix A. This present study was approved by the ethical review board of the Friedrich-Alexander-University Erlangen-Nürnberg (approval number: No. 3755 and 329_16B) in accordance with the Declaration of Helsinki. All patients gave written informed consent.

### 2.2. Immunohistochemistry (IHC)

IHC was performed using tissue microarray (TMA) sections (4 µm tissues) using Ventana BenchMark Ultra Stainer (Indianapolis, IN, USA) and Dako Link 48 (Dako, Twinsburg, OH, USA) autostainers in laboratories accredited according to DIN EN ISO/IEC 17020. Spatially organized TMAs covering the invasion front and tumor center were constructed as described previously [18,25,26]. Per patient, a total number of four TMA cores (two cores per invasion front and tumor center) with a diameter of 1.0 mm was obtained from donor FFPE blocks.

PD-L1 stainings were performed with closed CDAs as previously described (Appendix A) [18] and analyzed according to established scoring algorithms: (1) Ventana IC score, (2) CPS, and (3) TC_area_/IC_area_ score (Appendix A). PD-L1 stains were rated by two independent pathologists, who were blinded to clinical follow-up data at the time of analysis. To achieve consensus in ratings, strongly discrepant cases were rated by a third independent pathologist. All rating pathologists participated in multiple professional PD-L1 trainings for all analyzed PD-L1 scoring algorithms. Based on previously described interchangeability for TC (22c3, SP263, 28-8) and IC detection (all four assays), PD-L1 consensus scores (median expression across assays) were built to minimize inter-assay and section influences [18]: TC dependent consensus scores (CPS, TC_area_) excluding SP142 results due to decreased TC detection sensitivity; Ventana IC and IC_area_ consensus scores including the results of all four assays [18]. IHC staining using a comprehensive panel of immune cell phenotypes CD3, CD8, FOXP3, CD56, CD68, CTLA-4, LAG3, and PD-1 and the cytokine Granzyme B are summarized in Appendix A.

### 2.3. Digital Image Analysis Using QuPath

Stained slides were scanned on a slide scanner (P250 Flash, 3DHistech, Budapest, Hungary) and annotated by two pathologists (M.E., V.W.) in QuPath (https://qupath.github.io; RRID:SCR_018257, last access: 5 January 2021) [28]. TMA cores were quality controlled for the presence of tumor tissue and proper staining results. Cores with either absence of representative tumor tissue (complete absence, or less than 5% of the TMA core area) or staining artifacts (e.g., DAB shades) were excluded from the analysis. Total number of positive cells expressing CD3, CD8, FOXP3, CD56, CD68, CTLA-4, LAG3, PD-1, and GZMB were quantified per mm2 in all viable TMA cores via QuPath. For further analysis, median counts were log2 transformed. Log2-transformed values were Z-score-transformed to build a combined immune cell infiltration score (ICIS).

### 2.4. Assessment of Stromal Associated Tumor-Infiltrating Lymphocytes (sTILs)

sTILs were analyzed using hematoxylin and eosin-stained (HE) slides, as previously described [25,26]. Briefly, sTILs were scored as a semiquantitative percentage of mononuclear sTILs covering the desmoplastic tumor stroma. Necrotic or large fibrotic areas, papillary tumor parts, and carcinoma in situ were excluded [29].

### 2.5. Statistical Analysis

Disease-specific survival (DSS; time to disease-specific death) and disease-free survival (DFS; time to local recurrence and/or distant metastasis) curves were estimated by Kaplan–Meier regressions. Median potential follow-up was calculated by reverse Kaplan–Meier regression. Subgroup comparisons were carried out using nonparametric Mann–Whitney tests. Overlap of subgroups was visualized with Venn diagrams. Multivariable survival analyses were conducted using Cox proportional hazards regression modeling to assess the magnitude of impact (i.e., the hazard ratio [HR]) of different PD-L1 cut-offs and cluster groups, while adjusting for well-established clinical-pathological variables. All multivariable Cox regression analyses were adjusted for pT-Stage, pN-Stage, lymphovascular invasion (L), age, gender, resection margin status, receipt of an adjuvant platinum-containing chemotherapy regimen, and tumor grading. Multivariable models are summarized in Appendix A. Power analysis, randomization, and blinding were not performed (not applicable for present study type).

All *p*-values were two-sided, and *p*-values < 0.05 were considered significant. Cluster analyses were performed by unsupervised hierarchical clustering based on average link algorithm (WPGMA, weighted pair-group method using arithmetic averages) using Euclidean distance as metric scale. All statistical analyses were performed by GraphPad Prism 8.1.2 (GraphPad Software Inc., La Jolla, CA, USA; RRID:SCR_002798) and JMP SAS 13.2 (SAS, Cary, NC, USA).

## 3. Results

### 3.1. Result Text

#### 3.1.1. Clinically Applied PD-L1 Scoring Algorithms are Heterogeneous

We studied 193 cases of the CCC-EMN MIBC cohort. The distribution of gender (female patients: 54/193, 28%; male patients: 139/193, 72%) and age (mean age: 69.66 years) represents the common distribution in MIBC. Forty-nine patients received platinum-based adjuvant chemotherapy (49/193; 25.4%). Detailed patient characteristics are depicted in Appendix A. To analyze algorithm comparability, we performed PD-L1 staining with four approved PD-L1 CDAs and built consensus scores for each algorithm. A total of 103 cases were positive by at least one algorithm (53.3%; 103/193). Total positive tumors cases were 89 for TC_area_25%/IC_area_25%, 70 cases for Ventana IC5%, and 84 cases for CPS10. Of all positive cases, 52.4% (54/103) were positive for all three algorithms (Figure 1A), 20.4% (21/103) were exclusively positive for CPS10 and TC_area_25%/IC_area_25%, 9.7% exclusively for Ventana IC5% and TC_area_25%/IC_area_25% (10/103), and 1% exclusively for Ventana IC5% and CPS10 (1/103; Figure 1A). Each specific algorithm exclusively classified cases positive while the others did not: Ventana IC5%: 5/103 (4.9%); TC_area_25%/IC_area_25%: 4/103 (3.9%); CPS10: 8/103 (7.8%; Figure 1A). Inter-algorithm overall percentage agreement (OPA) ranged between 63.1% and 87.7% (Figure 1A). Figure 1B shows the distribution of PD-L1 scoring results in PD-L1 positive subsets of the algorithms. While IC5%, IC_area_25% and TC_area_25% positive tumors were enriched with PD-L1 positive IC or TC, CPS10 positive tumors represented a mixed continuum of tumors enriched with PD-L1 positive IC and TC (Figure 1B). In summary, current PD-L1 scoring algorithms identify very different tumor subsets with either predominant PD-L1 expression on IC (Ventana IC5%), TC (CPS10), or both (TC_area_25%/IC_area_25%).

#### 3.1.2. Scoring Algorithms Identify Tumors with Different Immune Cell Phenotypes and Correlate with Diverse Patient Outcomes

Based on the inter-algorithm variability, we correlated the algorithms with DSS and DFS to assess potential survival differences. CPS10 (Figure 2A), TC_area_25%/IC_area_25% (Figure 2B), and TC_area_25% did not associate with survival (Figure 2C). Patients with high PD-L1 expression on IC showed a trend for improved survival as assessed by IC_area_25% alone (Figure 2D; not significant in multivariable survival models) and significantly improved DSS (5-year DSS rate: 61%; HR_mult_ = 0.44 [95%-CI = 0.27–0.72]) and DFS (5-year DFS rate: 58%; HR_mult_ = 0.42 [95%-CI = 0.26–0.68]) as assessed by Ventana IC5% (Figure 2E).

General immune cell infiltration and immune checkpoint expression (ICIS) did not differ between the algorithms (Figure 3A). By applying unsupervised clustering of immune cell populations quantified at the invasion front and in the tumor center expressing CD3 (Pan-T-cells), CD8/GZMB (cytotoxic T-cells), FOXP3 (T_reg_-cells), CD56 (NK-cells), CD68 (macrophages), CTLA-4, LAG3, PD-1, and PD-L1 (PD-L1 consensus scores for IC and TC), we identified four different immunological tumor immune phenotypes, each correlating with different immune cell infiltration levels (ICIS) and patterns of PD-L1 expression (Figure 3B): (1) “Evasion” with high PD-L1 TC expression, low PD-L1 IC expression, and intermediate immune infiltration; (2) “Inflamed: High” with high PD-L1 IC expression, low PD-L1 TC expression, and high immune infiltration; (3) “Inflamed: Low” characterized by moderate immune infiltration, low PD-L1 IC, and mostly absent PD-L1 TC expression; and (4) “Uninflamed” with mostly absent inflammation and PD-L1 expression on IC and TC (Figure 3B,C; data distribution depicted in Appendix A). Patients with highly inflamed tumors and high PD-L1 IC expression (“Inflamed: High”; Figure 3B,C) showed significantly longer DSS (5-year DSS: 74.7%; Figure 3D) and DFS (5-year DFS: 75.1%;Figure 3E) in uni- and multivariable survival models compared to patients with tumors showing constitutive PD-L1 TC expression (“Evasion”; 5-year DSS: 29.8%; 5-year DFS: 25.8%), low inflammation and PD-L1 expression (“Inflamed: Low”; 5-year DSS: 43.1%; 5-year DFS: 43.7%), or tumors with mostly absent inflammation (“Uninflamed”; 5-year DSS: 27.7%; 5-year DFS: 23.7%;Figure 3D,E). Multivariable HRs are depicted below survival curves (Figure 3D,E), and full multivariable models are summarized in Appendix A. Representative images of HE morphology (sTILs), CD8^+^-infiltration, and PD-L1 expression on IC and/or TC of immune phenotypes are depicted in Figure 3F.

To further analyze if algorithms identify patients with different immune cell phenotypes and outcome, we constructed Venn diagrams of PD-L1 statuses and immunological tumor groups. While CPS10 included all “Evasion” tumors associated with worse outcomes (100%; 27/27), it missed 2 “PD-L1 IC high, Inflamed” cases that showed favorable survival (7.4%; 2/27; Figure 4A). Ventana IC5% preferentially identified “Inflamed: High” tumors with favorable outcomes (96.3%; 26/27), but missed 13 “Evasion” cases with worse survival rates (48.1%; 13/27; Figure 4B). TC_area_25%/IC_area_25% covered both clusters sufficiently by classifying 100% (27/27) of “Evasion” and 96.3% (26/27) of “Inflamed: High” tumors as PD-L1 positive (Figure 4C). In summary, current PD-L1 scoring algorithms identify “PD-L1” positive tumors, but are not able to accurately reflect immune cell phenotypes, thus mixing tumor groups with different prognostic and predictive potential.

#### 3.1.3. “PD-L1 Negative” Tumors with Present Anti-Tumoral Immune Responses Can be Identified by sTILs and Digital CD8^+^ T-Cell Scoring

Although moderate immune infiltration was present by ICIS, 48 tumors of the “Inflamed: Low” cluster were classified as PD-L1 negative by at least one algorithm: 46/48 cases (95.8%) by Ventana IC5%, 44/48 cases (91.7%) by TC_area_25%/IC_area_25%, and 48/48 cases (100.0%) by CPS10. Thirty-three of these cases were congruently classified PD-L1 negative by all three algorithms (68.8%). Digital CD8 (positivity cut-off = median: ≥201 CD8^+^ cells per mm^2^) and HE slide-based semiquantitative sTILs scoring (cut-off = median: ≥10% sTILs) sufficiently identified 39 (39/48; 81.3%) of these tumors with relevant anti-tumoral immune infiltrates, despite their PD-L1 negativity (Figure 5).

## 4. Discussion

Although PD-L1 testing is strictly required for first-line ICI of mUC, its context with immune cell phenotypes in the TIME is understudied [30]. In clinical trials PD-L1 is usually considered as a separate biomarker with dichotomized cut-offs (negative vs. positive), but without or only limited interpretation of its spatial or relative distribution (IC vs. TC) and expression with immune cell phenotypes and amounts of infiltrates. In addition, only little is known about the biological meaning of differential PD-L1 expression on TC and IC in the MIBC TIME in regards to immune therapy of naïve patients, and if these different tumor groups should be treated differently to achieve a better and more tailored patient response to immunotherapy. Recently, we described different MIBC immune cell phenotypes associated with different tumor subtypes, inflammation levels, PD-L1 distribution/expression, and patient outcomes [25,26]. Importantly, we discovered a specific inflamed phenotype with strong constitutive overexpression of PD-L1 on TC associated with poor survival while highly inflamed tumors with predominantly induced PD-L1 positivity on IC showed improved survival [25]. However, until today, no studies have been conducted to comprehensively characterize the immunological context of clinically applied PD-L1 scoring algorithms, their relation to immune cell phenotypes and, importantly, if they accurately reflect the baseline prognostic and predictive meaning of differential PD-L1 expression in tumors. Therefore, we conducted the present investigation: (1) to determine the inter-algorithm variability and the immunological context of different clinically applied PD-L1 scoring algorithms, (2) validate the baseline prognostic and potential predictive value of differential PD-L1 expression in relation to different immune cell phenotypes in immune therapy naïve patients, and (3) to test if the differential prognostic meaning of differential PD-L1 expression together with different immune cell phenotypes can be sufficiently reflected by current PD-L1 scoring algorithms.

Multiple drug-specific scoring algorithms are considered to substantially contribute to inconsistencies in PD-L1 scoring. As previously shown and presently validated on primary MIBC in this investigation, the inter-algorithm variability is substantial with OPAs, ranging from 63.1% to 87.7% [18]. This results in heterogeneous PD-L1 “positive” patient populations with important clinical implications. For example, while larger numbers of tumors are uniformly classified as PD-L1 positive or negative, a clinically relevant number of tumors shows discordant PD-L1 statuses leading to patient subsets, which we predict could exclusively receive pembrolizumab (29/103) or atezolizumab (present study: 15/103) despite the presence of high immune infiltration or PD-L1 expression. Importantly, this inter-algorithm variability is not exclusively explainable by different cut-offs, but also by divergent algorithm composition. While the Ventana IC5% algorithm preferentially identified tumors with high PD-L1 IC expression, thus showing the smallest overlap with mainly PD-L1 TC expression dependent CPS, the TC_area_25%/IC_area_25% algorithm overlaps significantly with both other algorithms due to the equal consideration of PD-L1 positive TCs and ICs. The sole assessment of PD-L1 with binary cut-offs detached from the biological context results in an oversimplification of complex biological processes occurring in the TIME. For example, constitutive PD-L1 TC expression found in “Evasion” phenotype tumors associate with a worse survival of immune therapy-naïve patients, thus indicating a potential benefit of ICI [25]. This particularly matches with observations from the KEYNOTE-045 trial, where patients with a CPS ≥ 10 receiving chemotherapy (control arm) showed worse survival than those with CPS < 10, which has been attributed to high PD-L1 TC expression [5]. Nevertheless, although TC-dependent algorithms such as CPS10 or TC_area_25%/IC_area_25% (TC_area_25% component) can identify these tumors with a poor baseline survival and a potential ICI benefit, they mix these cases with tumors with weak PD-L1 TC expression, strong PD-L1 IC expression, high overall immune infiltration, and favorable baseline survival (“Inflamed: High”). Thus, all these factors together may disguise the true predictive and prognostic potential of this specific immune phenotype. On the other hand, the Ventana IC5% and the IC_area_25% component of the TC_area_25%/IC_area_25% algorithm identified the majority of highly inflamed tumors with high PD-L1 IC expression (“Inflamed: High”) and an improved baseline prognosis. This matches with results from the IMvigor 211 trial, where patients with high PD-L1 IC expression showed increased survival rates compared to ” the intention to treat population” regardless of administered therapy (atezolizumab/chemotherapy). This supports a general positive prognostic influence of PD-L1 IC expression, as shown by our previous and present results [7,25]. Similar results have been reported from Study 1108, where patients with high PD-L1 IC expression also showed favorable outcomes [31]. Moreover, our present data are strongly supported by recent results from IMvigor010, where atezolizumab failed to prolong patient outcomes when administered as adjuvant maintenance therapy after radical cystectomy. Although not analyzed in detail, the authors reported that high PD-L1 expression on IC (IC ≥ 5%) indicated favorable post-cystectomy outcomes but was not able to predict benefits of adjuvant atezolizumab treatment [32]. In contrast, preliminary results from Checkmate274 showed that adjuvant nivolumab treatment following radical surgery prolongs disease-free survival partially dependent on PD-L1 expression on TC (NCT02632409). Another important issue is that PD-L1 assessment consistently fails to detect tumors with relevant immune infiltration, which frequently respond to ICI (“Inflamed: Low”) [33,34,35]. Here, we show that HE sTILs scoring and digital quantification of intratumoral CD8^+^ cytotoxic T-cells might be a suitable complementary tool to detect these tumors. As shown previously for head and neck squamous cell carcinomas, these methods can be implemented in pathological routine laboratories [35].

In summary, our present study shows that PD-L1 testing is heavily influenced by several important factors. First, currently applied scoring algorithms show huge inter-algorithm variability, thus leading to the selection of differently composed patient tumor populations, which are considered PD-L1 positive. More importantly, we showed that none of the investigated PD-L1 algorithms sufficiently cover specific immune cell phenotypes and differential PD-L1 expression on IC or TC. While these methodological differences are clinically important, our data, consistent with clinical trial results, point to a much larger problem: the lack of consideration of biological immune cell phenotypes. As clearly shown in our cohort of immunotherapy-naïve patients, patients with different immune phenotypes show extremely different clinical outcomes, which we reported previously and further validated in the present study using an enlarged study cohort [25]. Furthermore, these immune cell phenotypes cannot be resolved properly using current PD-L1 algorithms. Thus, questions have to be raised, such as which patients benefit from immunotherapy or only have a minor benefit, since treatment can also be accompanied by relatively strong adverse side effects, as reported in adjuvant therapy settings [32]. For example, patients with a strong anti-tumor immune response and favorable baseline outcomes might benefit from immune therapy by further unleashing the immune response in the MIBC TIME. Patients with strong PD-L1 overexpression on tumor cells and subsequent immunosuppression, on the other hand, represent a prime example of immunotherapy candidates due to their poor baseline prognosis. In contrast to MIBC, where this phenotype is rare, in lung cancer PD-L1 shows predominant and strong expression on tumor cells. Interestingly, this is supported by pivotal lung cancer trials where immune therapy response is mainly dependent on strong overexpression of PD-L1 on the tumor cell surface. Patients with less inflamed or uninflamed tumors, on the other hand, could benefit from combination therapies with drugs, which reignite the immune response in combination with ICI in order to induce a sufficient anti-tumor immune response, which is currently being explored in different types of cancers (NCT02816021, NCT03019003, NCT01928576, NCT02959437).

## 5. Conclusions

Taken together, we support the idea that consideration of the immunological state of the MIBC TIME in a more comprehensive way is necessary instead of the current approach using only PD-L1 testing with fixed cut-offs. Importantly, immunological interactions between immune cells themselves and also with tumor cells appear to be more challenging to not only understand the biology in the MIBC TIME but also appear more complex compared to other cancers, such as lung cancer. Therefore, we propose that it is imperative to stratify patients according to specific immune phenotypes in context with immune checkpoints such as PD-L1 in future studies. This type of patient/tumor assessment, we believe, will result in better control of opposing prognostic and potential predictive values of certain immune biomarkers, as well as clarifying which patients might benefit from immunotherapy.

## 6. Limitations

The present study is limited by its retrospective nature in light of the fact all patients were not treated with immune therapy, thus not allowing us to draw conclusions about the predictive value of the present findings. Although the PD-L1 assessment was carried out on spatially designed TMAs consisting of four tissue cores per patient, TMA analyses might not properly reflect the heterogeneity of PD-L1. Given that other studies reported negative prognostic effects of PD-L1 expression on IC, these results oppose our present results [36,37]. One explanation for this difference may be that both patient trials assessed PD-L1 as a static marker with very low cut-off values (1% positivity for TC and IC) without setting PD-L1 expression in its biological context [36,37]. Additionally, the study of Wang et al. investigated a broad range of bladder tumor stages consisting of non-invasive, low-grade tumors (mainly PD-L1 negative) to pT4 high-grade tumors with an abundance of PD-L1 positive TC and IC [36]. Thus, different patient outcomes stratified by PD-L1 assessment observed in this study were possibly biased due to comparison of a range of tumor stages with different baseline patient outcomes (e.g., pTa low grade tumors with favorable baseline prognosis and low/absent PD-L1 expression versus pT2–pT4 high-grade tumors with poor baseline survival and high PD-L1 expression). This is also reflected by the borderline significance of PD-L1 expression in multi-variable adjusted survival analyses in this study [36]. Taken together, it will be essential in the future for studies to validate our findings.

## Figures and Tables

**Figure 1 cancers-13-02327-f001:**
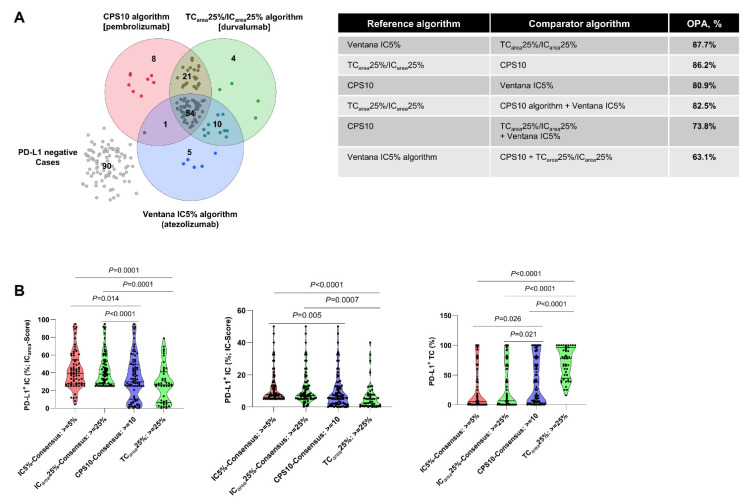
Agreement of different PD-L1 scoring algorithms and cut-off systems in urothelial cancer. (**A**) Left panel: Venn diagram showing the overlap of cases classified as PD-L1 positive according to the specific PD-L1 scoring algorithms. Right panel: overall percentage agreements (OPA) across different PD-L1 scoring algorithms. (**B**) Distribution of continuous consensus scoring values derived by the Ventana IC algorithm, TC_area_/IC_area_ algorithm, and combined positivity score (CPS) according to specific cut-off levels. Abbreviations: IC = immune cell(s); TC = tumor cell(s).

**Figure 2 cancers-13-02327-f002:**
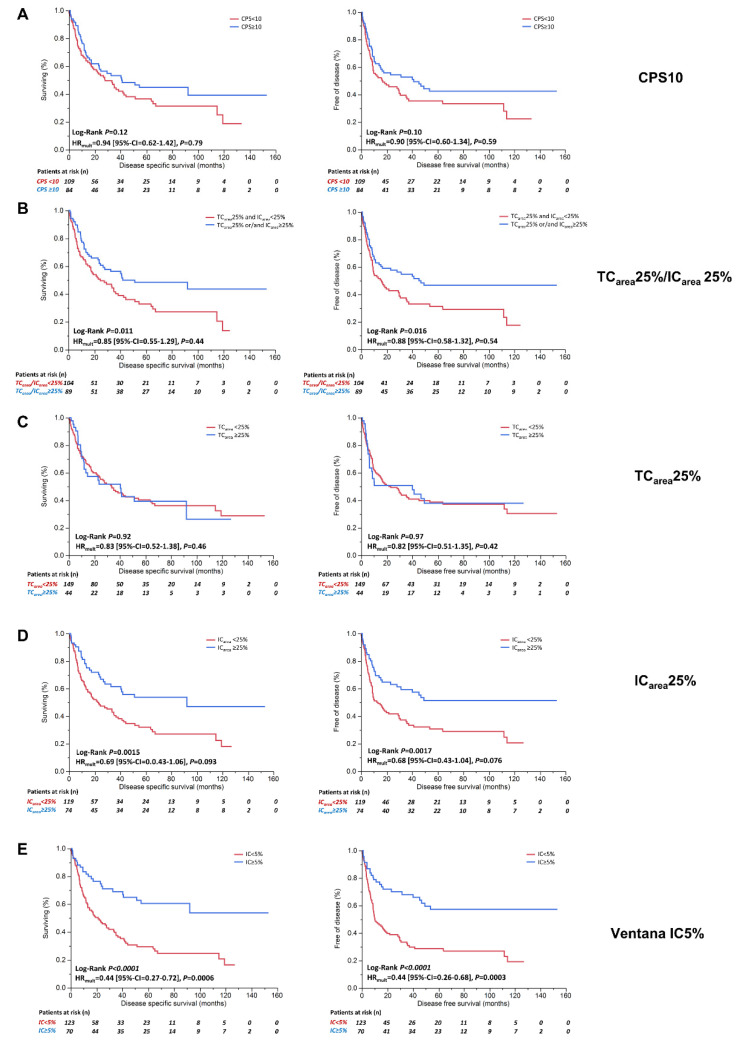
Prognostic impact of different PD-L1 scoring algorithms and cut-offs. Univariable Kaplan–Meier analyses of disease-specific survival (months) and disease-free survival (months) according to (**A**) CPS10, (**B**) TC_area_25%/IC_area_25%, (**C**) TC_area_25% alone, (**D**) IC_area_25% alone, and (**E**) Ventana IC5% algorithms. Univariable log-rank *p*-values and multivariable hazard ratios (HR) derived by Cox regression analyses are depicted below survival curves. Detailed multivariable survival models are summarized in Appendix A. All multivariable Cox regression analyses exploring the displayed indicator variables were adjusted for pT-Stage, pN-Stage, lymphovascular invasion (L), age, gender, resection margin status, receipt of an adjuvant platinum-containing chemotherapy regimen, and tumor grading (WHO2016 and WHO1973). Patients at risk at various milestones are depicted below survival curves. Abbreviations: CPS = combined positivity score; HR = Hazard Ratio; IC = immune cell(s); TC = tumor cell(s).

**Figure 3 cancers-13-02327-f003:**
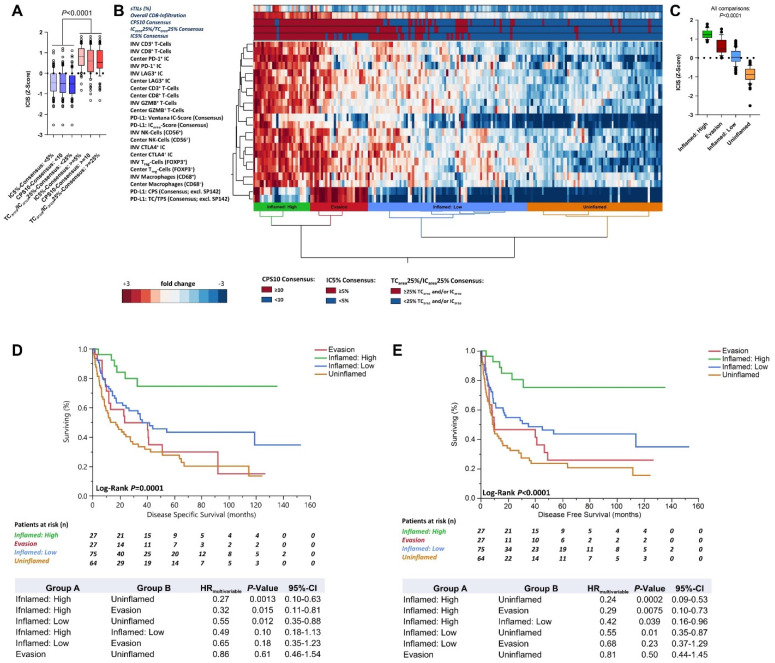
Immune biological correlates of PD-L1 scoring algorithms. (**A**) Overall immune infiltration (immune cell infiltration score; ICIS) according to PD-L1 scoring algorithms and respective cut-offs for PD-L1 negativity and positivity. (**B**) Unsupervised hierarchical cluster analysis of individual immune cell populations estimated at the invasion front and tumor center together with PD-L1 consensus scoring results (CPS, Ventana IC score, TPS/TC_area_ score, IC_area_ score) reveals four immune biological cluster groups: “Inflamed: High”, “Evasion”, “Inflamed: Low”, and “Uninflamed”. Stromal tumor-infiltrating lymphocyte count, overall CD8^+^-infiltration, and PD-L1 classification results according to CPS10, Ventana IC5%, and TC_area_25%/IC_area_25% are depicted above the heat map in H-bar plots. (**C**) General immune infiltration (ICIS) according to cluster groups. Disease-specific (**D**) and disease-free survival (**E**) according to cluster groups. Univariable log-rank *p*-values and multivariable hazard ratios (HR) derived by Cox regression analyses are depicted below survival curves. Detailed multivariable survival models are summarized in Appendix A. All multivariable Cox regression analyses exploring the displayed indicator variables were adjusted for pT-Stage, pN-Stage, lymphovascular invasion (L), age, gender, resection margin status, receipt of an adjuvant platinum-containing chemotherapy regimen, and tumor grading (WHO2016 and WHO1973). Patients at risk at various milestones are depicted below survival curves. (**F**) Representative images of HE morphology, CD8^+^-infiltration, and PD-L1 expression on IC or TC according to the four cluster groups; TMA core images were taken in 50× magnification, magnified snapshots were taken in 200× magnification. Abbreviations: CPS = combined positivity score; HR = hazard ratio; IC = immune cell (s); ICIS = immune cell infiltration score; TC = tumor cell (s).

**Figure 4 cancers-13-02327-f004:**
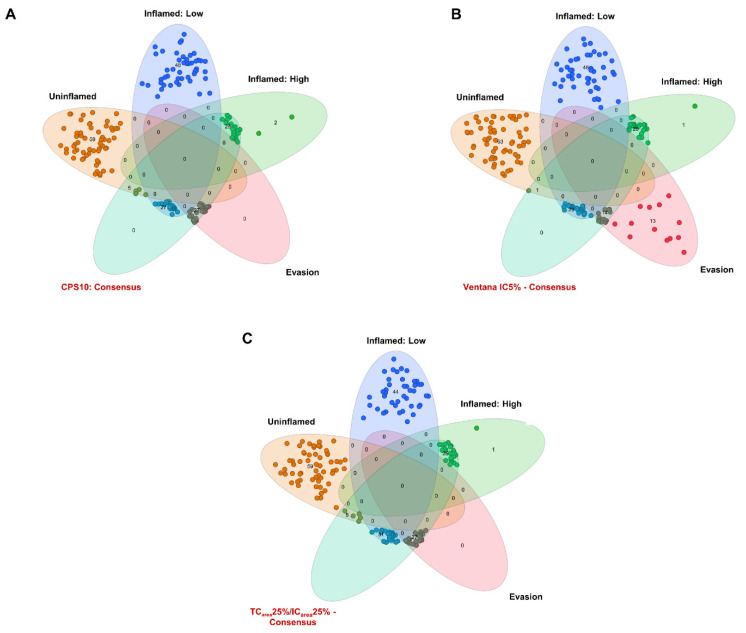
Correlation of PD-L1 scoring algorithms with immune phenotypes. Venn diagrams were constructed to illustrate overlaps of patients classified as positive by the CPS10 (**A**), the Ventana IC5% algorithm (**B**), the TC_area_25%/IC_area_25% algorithm (**C**) with the four immune biological cluster groups (“Inflamed: High”, “Evasion”, “Inflamed: Low”, “Uninflamed”). Abbreviations: CPS = combined positivity score; IC = immune cell (s); TC = tumor cell (s).

**Figure 5 cancers-13-02327-f005:**
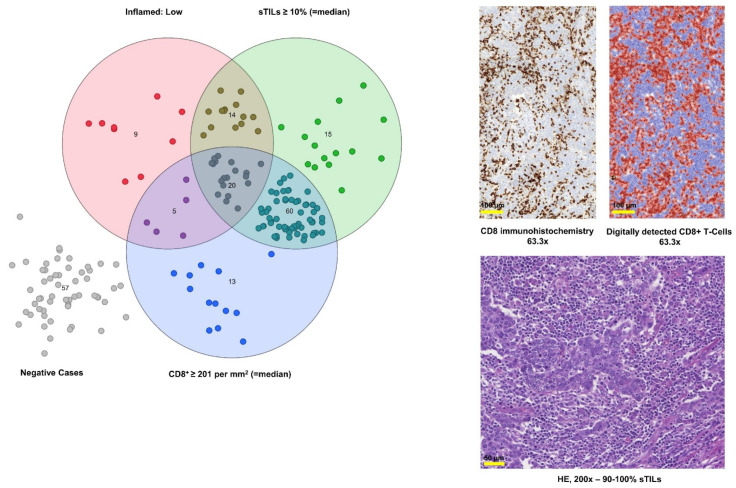
Scoring of sTILs and digital quantification of tumor infiltrating CD8^+^ T-cells detected in inflamed tumors with low PD-L1 expression (“Inflamed: Low”). The Venn diagram illustrates the overlap of “Inflamed: Low” tumors with moderate immune infiltration and sTILs scoring on HE slides as well as digital CD8^+^ quantification. Images on the right illustrate digital CD8^+^ cytotoxic T-cell scoring (upper panel; 63.3× magnification) and of sTILs scoring (lower picture; HE 200× magnification). Abbreviations: HE = hematoxylin & eosin; sTILs = stromal tumor infiltrating lymphocytes.

## Data Availability

Data from the CCC-EMN cohort can be shared on request in anonymized data format with an additional allowance of the German authority for clinical data security.

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
