# Peer review of "Integration of Spatial PD-L1 Expression with the Tumor Immune Microenvironment Outperforms Standard PD-L1 Scoring in Outcome Prediction of Urothelial Cancer Patients"

_cancers, 2021, doi:10.3390/cancers13102327_

Round 1
Reviewer 1 Report
The present work is well described and articulated.Author Response
Comments and Suggestions for Authors
The present work is well described and articulated.
We are very grateful for your judge on our present study.
Reviewer 2 Report
In this study, the authors compare different PD-L1 algorithms (CPS, TCarea25%/ICarea25%, etc), which is a very relevant topic as the use of different algorithms in studies leads to conflicting results about the prognostic/predictive value of PD-L1. Nevertheless, what makes this study of limited value is that the patients in this study were not treated with checkpoint inhibitors and that there is considerable overlap with previously published research (ref 18/25).
Major points
- In this study, the authors use a PD-L1 consensus scores (median expression across four assays), based on previously reported interchangeability between these assays. As the main focus of this article is comparing the different PD-L1 assays and the authors have performed all four assays, it would be nice if the authors could confirm previously published results and add a figure/table, possibly as supplement, showing the strong agreement between the different assays.
- Whereas the comparison between the different PD-L1 algorithms is new, there is considerable overlap with a previous study of this research group (ref 25). In this paper, the authors describe 4 comparable clusters with better outcomes for the PD-L1 IC, inflamed group (there called the inflamed high-group).
- The patients in this cohort were not treated with checkpoint inhibitors, making it impossible to compare predictive value for response to checkpoint inhibitors between assays
- The authors conclude that comprehensive assessment of the TIME provides more information than PD-L1 testing alone. In the method section, the authors describe that they performed comprehensive IHC analysis in the entire cohort, incl CD3, CD8, FoxP3, CD68 etc. Nevertheless, they do not evaluate these markers individually and seem to only use a combined immune cell infiltration score when performing unsupervised clustering. I would suggest to also analyze these markers individually, i.e. perform unsupervised clustering with all these individual cell subsets to gain more insight into immune subtypes/relevant prognostic markers. This might also give the study more novelty.
Minor comments:
Abstract:
- The abstract could be improved. The method section is a bit unclear and the results section is difficult to read
Introduction
- Language/style changes are needed, e.g.
- Add a full stop between “regimens” and “however” (l. 58)
- “changed the first cis-platinum ineligible patients and second-line treatment options”(l.61-62)
- “durable long term” change in to “durable” or “long-term” (l.63)
- I wouldn’t say that patients demonstrate low ORRs to chemotherapy (l. 59). ORR to platinum-based chemotherapy are not low
- In both the IMvigor211 (atezolizumab) and Keynote-045 (pembrolizumab), PD-L1 did not appear to be a strong predictive biomarker (l. 71-73)
Methods
- Please describe number and size of TMA cores.
- If more than 1 core was used per algorithm it might be interesting to also describe spatial heterogeneity of PD-L1 expression..
- Please describe which parameters were used for unsupervised clustering.
Results
- Paragraph 3.1 is called ”result text”
- In figure 1B: how is it possible that for example some patients with an "ICarea25% >= 25%" have a ICarea score < 25?
- It is nice to see how well the different algorithms overlap with the different clusters in figure 4, but I do wonder what the added value of the Kaplan Meier curves in figure 4 is.
Discussion
- 312-314: I would remove “predictive value”. As patient were not treated with checkpoint inhibitors it is not possible to say something about the predictive value of the different algorithms in this study. In addition, I think that point 2 is not new compared to ref 25, even though the cohort is now slightly bigger.
- Although the authors find PD-L1 expression on ICs to be related with favorable outcomes in this study, there is also data suggesting the opposite. The authors are now only referring to studies that support a link with positive outcome. I would like the authors to also comment on studies showing a negative association, e.g.
- DOI: 10.1111/cas.13887 à both PD-L1 IC and TC related with poor prognosis
- DOI: 10.1016/j.clgc.2018.05.015 à PD-L1+ IC related with shorter RFS
Author Response
In this study, the authors compare different PD-L1 algorithms (CPS, TCarea25%/ICarea25%, etc), which is a very relevant topic as the use of different algorithms in studies leads to conflicting results about the prognostic/predictive value of PD-L1. Nevertheless, what makes this study of limited value is that the patients in this study were not treated with checkpoint inhibitors and that there is considerable overlap with previously published research (ref 18/25).
Major points
In this study, the authors use a PD-L1 consensus scores (median expression across four assays), based on previously reported interchangeability between these assays. As the main focus of this article is comparing the different PD-L1 assays and the authors have performed all four assays, it would be nice if the authors could confirm previously published results and add a figure/table, possibly as supplement, showing the strong agreement between the different assays.
We really appreciate this very important comment. The PD-L1 data used for generating the consensus score for each scoring algorithm were already published in our past study comparing the comparability of the different assays (Eckstein et al, 2019, European Journal of Cancer). All cases presented in the present manuscript were included in this previous study. Unfortunately, the remaining cases in the “EJC” paper, which stemmed from another pathological institute, were not available for the present study. However, at this point it is very important to clarify, that the main purpose of the present study was to compare the different PD-L1 scoring algorithms, and not the different assays themselves. Although the different assays show wide interchangeability (with the exception of SP142), minor discrepancies remain between the different stainings. Therefore, we decided to build the PD-L1 consensus score (as outlined in the M&M section) to fully focus on the differences between the scoring algorithms independent of inter-assay variabilities (CPS vs. IC-Score vs. durvalumab ICarea/TCarea-Score).
Whereas the comparison between the different PD-L1 algorithms is new, there is considerable overlap with a previous study of this research group (ref 25). In this paper, the authors describe 4 comparable clusters with better outcomes for the PD-L1 IC, inflamed group (there called the inflamed high-group). The authors conclude that comprehensive assessment of the TIME provides more information than PD-L1 testing alone. In the method section, the authors describe that they performed comprehensive IHC analysis in the entire cohort, incl CD3, CD8, FoxP3, CD68 etc. Nevertheless, they do not evaluate these markers individually and seem to only use a combined immune cell infiltration score when performing unsupervised clustering. I would suggest to also analyze these markers individually, i.e. perform unsupervised clustering with all these individual cell subsets to gain more insight into immune subtypes/relevant prognostic markers. This might also give the study more novelty.
This is a very important point and idea from Reviewer 2. Therefore, we decided to repeat and to enlarge different analyses to bring the present results in line with our previous study (Pfannstiel et al, 2019, CIR). As suggested, we performed unsupervised hierarchical clustering of all individual markers including the PD-L1 variables to set the present study results into a deeper immune biological context and to provide confirmatory analyses of previously published results with a broader immune cell panel. In addition, we termed the cluster groups according to our previously published study. All Venn diagrams in Figure 4 and Figure 5 were remodeled to reflect the correct overlap with the different PD-L1 cut-off groups and the new cluster groups/immune phenotypes. All multivariable survival models including the immune phenotypes were repeated and are now displayed in the respective updated and revised supplementary table. All other sections of the manuscript were updated accordingly.
The patients in this cohort were not treated with checkpoint inhibitors, making it impossible to compare predictive value for response to checkpoint inhibitors between assays.
We absolutely agree with this major concern. Thus, we added a limitation section at the end of the paper clearly stating this limitation. The discussion already refers to this issue multiple times. However, we believe that the present results are important to underline that current PD-L1 scoring algorithms are not sufficient to cover the intrinsic immune biology of muscle-invasive urothelial cancer. Although we are not able to demonstrate directly that this is important for response prediction, our analyses clearly show that the intrinsic immune biology of muscle-invasive urothelial carcinoma is extremely important for patient outcomes, and can’t be reflected by PD-L1 scoring algorithms at all. This is especially important, since systematic, independent and critical analyses on the impact of immune phenotypes in relation to currently applied PD-L1 scoring algorithms are completely lacking. The main contribution of our study to the field is, that we are able to demonstrate that PD-L1 scoring critically fails to resolve immune biology, thus leading to a very problematic mixing of patients with completely different immune phenotypes and outcomes in one “PD-L1 positive” pot.
Minor comments:
Abstract:
The abstract could be improved. The method section is a bit unclear and the results section is difficult to read.
The abstract has been rephrased to clarify the method section and to improve the readability of the results part.
Introduction:
Language/style changes are needed, e.g.
Add a full stop between “regimens” and “however” (l. 58) & I wouldn’t say that patients demonstrate low ORRs to chemotherapy (l. 59). ORR to platinum-based chemotherapy are not low.
The sentence “[…] However, patients demonstrate low objective response rates (ORR). […]” has been deleted from the introduction.
“changed the first cis-platinum ineligible patients and second-line treatment options”(l.61-62)
This has been corrected and highlighted in the revised version: […] changed the first- and second-line treatment options […]
“durable long term” change in to “durable” or “long-term” (l.63)
This has been changed to “long term”.
In both the IMvigor211 (atezolizumab) and Keynote-045 (pembrolizumab), PD-L1 did not appear to be a strong predictive biomarker (l. 71-73)
To correct this statement the sentences has been rephrased to “[…] However, in mUC predictive and prognostic value of PD-L1 is inconsistent between therapy lines (front-line vs. second-line) and different drugs (e.g., atezolizumab vs. pembrolizumab) […]”. In addition, relevant references of trials where PD-L1 seemed to predict response and to stratify outcomes were added to set the sentence in its right context.
Methods
Please describe number and size of TMA cores.
We added a clarifying statement: […] Per patient a total number of four TMA cores (two cores per invasion front and tumor center) with a diameter of 1.0 mm was obtained from donor FFPE blocks. […]
If more than 1 core was used per algorithm it might be interesting to also describe spatial heterogeneity of PD-L1 expression.
This is a very good point. However, this issue has been addressed in our past study published in “EJC (Eckstein et al, 2019)” already, where a substantial inter-core heterogeneity could be found. This matches with the overall high frequency of heterogeneous PD-L1 expression results which can be found in whole slides stains.
Please describe which parameters were used for unsupervised clustering.
Variables used for hierarchical clustering are now correctly displayed on the left side of the cluster and mentioned in the results section.
Results
Paragraph 3.1 is called ”result text”
“3.1. result text” refers to the mdpi standard template. “3.2 Figures, Tables and Schemes” is the second subheading, which is pre-specified by this template, and can be found on page 7 of 17.
In figure 1B: how is it possible that for example some patients with an "ICarea25% >= 25%" have a ICarea score < 25?
This is a very good point. There was a data display error in the left two panels (ICarea/IC-Score). Figure 1B is now adjusted accordingly and data should be displayed correctly.
It is nice to see how well the different algorithms overlap with the different clusters in figure 4, but I do wonder what the added value of the Kaplan Meier curves in figure 4 is.
This is great comment, and indeed the survival analyses do not add additional information. Moreover, they might be even more confusing than helpful. Therefore, we decided to remove these survival analyses. Figure 4 has been updated accordingly.
Discussion
312-314: I would remove “predictive value”. As patient were not treated with checkpoint inhibitors it is not possible to say something about the predictive value of the different algorithms in this study. In addition, I think that point 2 is not new compared to ref 25, even though the cohort is now slightly bigger.
The sentence has been rephrased. We replaced “determine” with “validate” to express that the present analysis is rather of confirmatory than of exploratory nature. In addition we removed “predictive” as suggested: […] Therefore, we conducted the present investigation: 1) to determine the inter-algorithm variability and the immunological context of different clinically applied PD-L1 scoring algorithms; 2) validate the baseline prognostic and potential predictive value of differential PD-L1 expression in relation to different immune cell phenotypes in immune therapy naïve patients; and 3) to test if the differential prognostic meaning of differential PD-L1 expression together with different immune cell phenotypes can be reflected by current PD-L1 scoring algorithms sufficiently. […] In addition we added a limitations section at the end of the manuscript, and stated that patients were not treated with immune therapy.
Although the authors find PD-L1 expression on ICs to be related with favorable outcomes in this study, there is also data suggesting the opposite. The authors are now only referring to studies that support a link with positive outcome. I would like the authors to also comment on studies showing a negative association, e.g.
DOI: 10.1111/cas.13887 à both PD-L1 IC and TC related with poor prognosis
DOI: 10.1016/j.clgc.2018.05.015 à PD-L1+ IC related with shorter RFS
Thank you for this advice. We absolutely agree that there are studies with opposing trends. Since these studies are not fitting to the context of the discussion we implemented another statement in the newly added limitations section that other studies reported opposing trends: […] At the background that other studies reported negative prognostic effects of PD-L1 expression on IC (opposing to our present results) (36, 37), external validation of our findings is needed. […]
However, we want to clarify that especially the above mentioned trials are different to the approach we applied in our analyses. Some pinpoints are important to mention here:
- PD-L1 expression in these two trials was assessed as a static marker with very low cut-off values (1% for TC and IC positivity)
- In the opposite, our results base on a dynamic and agnostic bioinformatical approximation of intrinsic phenotypes and inter-relation of multiple TIME markers. Other groups using these approaches found comparable results as we did (Sjödahl et al, Scientific Reports; Sharma et al, PNAS; Nassif et al, Cancers; Taber et al, Nature Communications).
- The Study of Wang et al. is difficult to interpretate as it investigates a broad range of different bladder tumor stages. Many studies in the past clearly showed that NMIBC and MIBC do not only show very different outcomes but also are biologically different entities. Of the investigated samples more than fifty percent represented “pTa-pT1” tumors. A further differentiation is not provided. However, as pT1 carcinomas and MIBC are “high grade” per definition of the WHO classification of 2004/2016 the number of 72 low grade carcinomas in this study suggests that the number of pTa tumors amounted to approximately 1/3 of the whole analyzed cohort. As shown by the authors, low grade tumors and “pTa-pT1” carcinomas were predominantly PD-L1 negative, while the included MIBC, especially those with positive nodal status were predominantly PD-L1 IC and TC positive. As pTa and pT1 carcinomas usually show overall survival rates of >85%, MIBC without nodal metastasis appr. 40-60%, and MIBC with present lymph node metastasis <15%, the results of the survival analyses are not surprising. This is further reflected in the multivariable survival analyses where only PD-L1 IC expressions and independent predictor for poor survival, but only with 95%-CIs nearly hitting 1. The strong survival differences between TC positive and negative cases in Kaplan-Meier regression not qualifying as an independent predictor for poor OS or RFS further underline the problematic mixing of pTa, pT1 and pT2+ carcinomas.
- The study of Pichler et al is also biased by exclusively including patients with pT-stages of pT3 or higher OR/and nodal metastases (= ultra high-risk subgroup of MIBC). This means that patients with stage pT2a/b and pN0 were completely excluded from the analyses, which represent appr. 30% of all MIBC patients. Statements of how the cohort was collected (selection vs. consecutively/population based) are lacking in this study.
Reviewer 3 Report
I would like to thank the authors for submitting this well thought-out and methodologically appealing manuscript. The topic is highly relevant in urology, the large variability in PD-L1 scoring is a relevant problem as it influences therapy. Further research is urgently required.
In my opinion, the present manuscript can significantly contribute to this topic and is therefore publishable after a few adjustments in the sense of a minor revision.
Introduction:
- Please introduce the TIME abbreviation again in the introduction.
Methods:
- Please state the time period in which the cystectomies were performed.
- Parts of the patient characteristics (e.g. sex, age) are already described in the methods section. Please report these in the results section instead (analogous to e.g. the recommendations of the Equator Network).
- I understand that power analysis and randomization were not performed. However, blinding of the pathologists, especially with regard to the clinical outcome, would have been possible and desirable. Has this been done? If not, this could be seen as a limitation / shortcoming of the study.
- How does the PD-L1 scoring on TMAs performs in comparison to the PD-L1 scoring on whole slide images? This might need to be mentioned as a short coming (please also see my comment n. 2 in the discussion part)
Results:
overall good.
- The visualisation could be further improved by providing representative PD-L1 stained images for the cluster subgroups with the corresponding TIL-HE slide (consider provision in the appendix / supplementary).
Discussion:
- No limitations are stated. Please re-evaluate your study carefully in regard to possible shortcomings and provide them accordingly.
- You do not discuss the fact that in your study PD-L1 scoring was performed on TMAs. Please explain whether this approach is equivalent to PD-L1 scoring in the routine setting where the whole slide is assessed.
Author Response
I would like to thank the authors for submitting this well thought-out and methodologically appealing manuscript. The topic is highly relevant in urology, the large variability in PD-L1 scoring is a relevant problem as it influences therapy. Further research is urgently required. In my opinion, the present manuscript can significantly contribute to this topic and is therefore publishable after a few adjustments in the sense of a minor revision.
We thank you for your review of our manuscript and appreciate your comments regarding our present study. All raised points were addressed and are answered point by point below.
Introduction:
- Please introduce the TIME abbreviation again in the introduction.
The abbreviation is now introduced at its first appearance within the introduction section.
Methods:
- Please state the time period in which the cystectomies were performed.
The time period is now stated in the material and methods section (“between 2004-2016”).
- Parts of the patient characteristics (e.g. sex, age) are already described in the methods section. Please report these in the results section instead (analogous to e.g. the recommendations of the Equator Network).
We removed the description of patient characteristics from the material and methods sections, and inserted them in the beginning of the results part instead. The new passage is highlighted in the revised manuscript.
- I understand that power analysis and randomization were not performed. However, blinding of the pathologists, especially with regard to the clinical outcome, would have been possible and desirable. Has this been done? If not, this could be seen as a limitation / shortcoming of the study.
This is a very important point. Indeed, basic data for building the combined PD-L1 consensus scores were obtained in 2018 by two independent pathologists which were blinded to the follow-up/clinical data at the time of scoring. Cases with strongly discrepant scoring results causing cut-off deviations from clinical important cut-offs were independently reviewed by a third pathologist. Afterwards consensus scores were achieved based on the third independent review. To clarify this we added a statement in the material and methods section: […] PD-L1 stains were rated by two independent pathologists, which were blinded to clinical follow-up data at the time of analysis. To achieve consensus rating, strongly discrepant cases were rated by a third independent pathologist. All rating pathologists participated in multiple professional PD-L1 trainings for all analysed PD-L1 scoring algorithms. […]
- How does the PD-L1 scoring on TMAs performs in comparison to the PD-L1 scoring on whole slide images? This might need to be mentioned as a short coming (please also see my comment n. 2 in the discussion part).
Heterogeneity of PD-L1 expression is a common issue in routine diagnostics. Therefore, this comment is highly relevant. As shown in our previous study (Eckstein et al, 2018, EJC), inter-core variability was substantial, thus at least partly reflecting PD-L1 heterogeneity. A total of four TMA cores from the invasion front and the tumor center (each 1.0 mm diameter, appr. 1mm2 tissue area, = 4 mm2 tumor tissue) were analyzed. This is now also stated in the material and methods section. Although this spatial design with four cores might reflect the heterogeneity of PD-L1 expression, we additionally added this to a new limitations section at the end of the manuscript.
[…] 6. Limitations
The present study is limited by its retrospective nature in light of the fact all patients were not treated with immune therapy, thus not allowing to draw conclusions about the predictive value of the present findings. Although PD-L1 assessment was carried out on spatially designed TMAs consisting of four tissue cores per patient, TMA analyses might not reflect heterogeneity of PD-L1 properly. […]
Results:
overall good.
- The visualisation could be further improved by providing representative PD-L1 stained images for the cluster subgroups with the corresponding TIL-HE slide (consider provision in the appendix / supplementary).
We updated Figure 3 accordingly. Representative images (HE, CD3, PD-L1) of each immune phenotype are now displayed in Figure 3F.
Discussion:
- No limitations are stated. Please re-evaluate your study carefully in regard to possible shortcomings and provide them accordingly.
This is a great suggestion. As mentioned above a limitation sections has now been now added at the end of the manuscript.
- You do not discuss the fact that in your study PD-L1 scoring was performed on TMAs. Please explain whether this approach is equivalent to PD-L1 scoring in the routine setting where the whole slide is assessed.
Copied from point 4 “results”: Heterogeneity of PD-L1 expression is a common issue in routine diagnostics. Therefore, this comment is highly relevant. As shown in our previous study (Eckstein et al, 2018, EJC), inter-core variability was substantial, thus at least partly reflecting PD-L1 heterogeneity. A total of four TMA cores from the invasion front and the tumor center (each 1.0 mm diameter, appr. 1mm2 tissue area, = 4 mm2 tumor tissue) were analyzed. This is now also stated in the material and methods section. Although this spatial design with four cores might reflect the heterogeneity of PD-L1 expression, we additionally added this to a new limitations section at the end of the manuscript.
[…] 6. Limitations
The present study is limited by its retrospective nature in light of the fact all patients were not treated with immune therapy, thus not allowing to draw conclusions about the predictive value of the present findings. Although PD-L1 assessment was carried out on spatially designed TMAs consisting of four tissue cores per patient, TMA analyses might not reflect heterogeneity of PD-L1 properly. […]
Round 2
Reviewer 2 Report
I thank the authors for their comprehensive reply. I agree that there is no need to report on the interchangeability between the different assays or intercore heterogeneity again if it already published elsewhere. My other major concerns are sufficiently answered.
Some minor suggestions:
- In their comments the authors write that the sentence “[…] However, patients demonstrate low objective response rates (ORR). […]” has been deleted from the introduction.(line 57-58), but it is still there.
- I appreciate that the authors took up a section on limitations in their manuscript. However the authors might elaborate a little bit more on the limitation in line 417-419 (about other studies reporting negative prognostic effects of PD-L1 expression on IC). They sent me a very comprehensive reply explaining the differences with their own study, maybe they can summarize this in 1-2 sentences and add it to the manuscript to make it a bit more stronger.
- Finally, I would suggest a final spell check (e.g. “a systematic analysis [..] IS lacking” (line 85-86)) and attention to the use of commas (e.g. “However, in mUC, predictive [..]” (line 70) and “ [..] atezolizumab / pembrolizumab monotherapy, ICI was restricted to patients [..]” (line74), “To analyze algorithm comparability, we performed [..]” (line 183))
Besides these small comments, I think it is a nice manuscript and I would suggest it is accepted for publication.
Author Response
I thank the authors for their comprehensive reply. I agree that there is no need to report on the interchangeability between the different assays or intercore heterogeneity again if it already published elsewhere. My other major concerns are sufficiently answered.
Thank you so much for your great comments. We implemented all changes as outlined in the following point by point reply:
Some minor suggestions:
In their comments the authors write that the sentence “[…] However, patients demonstrate low objective response rates (ORR). […]” has been deleted from the introduction.(line 57-58), but it is still there.
We appreciate this comment and we double checked it and deleted the sentence.
I appreciate that the authors took up a section on limitations in their manuscript. However the authors might elaborate a little bit more on the limitation in line 417-419 (about other studies reporting negative prognostic effects of PD-L1 expression on IC). They sent me a very comprehensive reply explaining the differences with their own study, maybe they can summarize this in 1-2 sentences and add it to the manuscript to make it a bit more stronger.
We added a summary of our previous comprehensive reply as follows among the manuscript and the limitation section:
[…]At the background that other studies reported negative prognostic effects of PD-L1 expression on IC, these results opposes our present results (36, 37). One explanation for this difference may be that both patient trials assessed PD-L1 as a static marker with very low cut-off values (1% positivity for TC and IC) without setting PD-L1 expression in its biological context (36, 37). Additionally, the study of Wang et al investigated a broad range of bladder tumor stages consisting of non-invasive low grade tumors (mainly PD-L1 negative) to pT4 high grade tumors with an abundance of PD-L1 positive TC and IC (36). Thus, different patient outcomes stratified by PD-L1 assessment observed in this study were possibly biased due to comparison of a range of tumor stages with different baseline patient outcomes (e.g., pTa low grade tumors with favorable baseline prognosis and low/absent PD-L1 expression versus pT2-pT4 high grade tumors with poor baseline survival and high PD-L1 expression). This is also reflected by the borderline significance of PD-L1 expression in multi-variable adjusted survival analyses in this study (36). Taken together, it will be essential in the future for studies to validate our findings. […]
Finally, I would suggest a final spell check (e.g. “a systematic analysis [..] IS lacking” (line 85-86)) and attention to the use of commas (e.g. “However, in mUC, predictive [..]” (line 70) and “ [..] atezolizumab / pembrolizumab monotherapy, ICI was restricted to patients [..]” (line74), “To analyze algorithm comparability, we performed [..]” (line 183))
We thank the reviewer for this comment and a native speaker and co-author (Pamela Strissel) of the manuscript further edited the manuscript for spelling and sentence structure.
Besides these small comments, I think it is a nice manuscript and I would suggest it is accepted for publication.
We thank the reviewer for the positive support and we believe the comments have greatly improved our manuscript.